# Carpal Tunnel Syndrome: A National Survey to Monitor Knowledge and Operating Methods

**DOI:** 10.3390/ijerph18041995

**Published:** 2021-02-18

**Authors:** Valentina Scalise, Fabrizio Brindisino, Leonardo Pellicciari, Silvia Minnucci, Francesca Bonetti

**Affiliations:** 1Department of Clinical Sciences and Translational Medicine, University of Rome Tor Vergata, I-00133 Rome, Italy; scalise.valentina@me.com (V.S.); fabrindi@gmail.com (F.B.); silviaminnucci8@gmail.com (S.M.); fra.bonetti@me.com (F.B.); 2Department of Medicine and Health Science “Vincenzo Tiberio”, University of Molise, I-86100 Campobasso, Italy; 3Neurorehabilitation Research Laboratory, IRCCS San Raffaele Pisana, I-00166 Rome, Italy

**Keywords:** carpal tunnel syndrome, Italian survey, physiotherapy, care management, rehabilitation

## Abstract

The aim of this article was to investigate the knowledge, management, and clinical practice of Italian physiotherapists concerning patients with carpal tunnel syndrome (CTS). A national cross-sectional survey consisted of 24 questions was administered from December 2019 until February 2020. A Chi-squared independence test was run to study any difference between subgroups of the sample and responses to the questionnaire. Five hundred and eight respondents completed the survey. Most respondents (*n* = 225/508; 44.3%) are under 29 years old, female (*n* = 256/508; 50.4%) and have been working as physiotherapists for less than 5 years (*n* = 213/508; 41.9%). Most of respondents correctly knows about the cause (*n* = 455/508, 89.6%), main signs and symptoms of CTS (*n* = 415/508, 81.70%) and administer education, manual therapy, myofascial techniques and therapeutic exercises (*n* = 457/508, 89.88%). Three hundred and sixty-four (71.68%) respondents were aware of the influence of psychosocial factors on the patient’s outcomes. The survey showed greater adherence to evidences by physiotherapists holding a master’s degree. The results are mostly comparable with other surveys structured all over the world on the same topic. Italian physiotherapists management of the CTS was not always in line with current evidence. Interventions such as education, manual therapy, therapeutic exercise, nerve and tendon glide techniques are widely used, while the orthotic is only offered by half of the sample.

## 1. Introduction

Carpal Tunnel Syndrome (CTS) is described as the most common compressive neuropathy. The prevalence in the American population, regardless of job type, is 7.8% [1]. In the general population, the incidence is 23% when the diagnosis is made with clinical and electrodiagnostic criteria, and yet the incidence is higher in the working population than in the general population [2]. In Italy, CTS is the most frequent of the pathologies affecting those who perform manual work, with a substantial increase from 2006 to 2010 (from 1731 to 4819) or an increase of over 170% [3].

The complaints of occupational disease registered by INAIL-National Institute of Accident Insurance-in the first five months of 2019 were 27,385,372 more than in the same period of 2018 (+1.4%). Among these the pathologies of the osteo-muscular system and of the connective tissue (15,556 cases) and of the nervous system (2741, with a superiority of the CTS) continue to represent the first professional diseases reported [4]. CTS determines direct and indirect costs for the entire socio-health system, but in particular when the ability to work is compromised [2,5,6]. The risk factors related to CTS are frequently identified in obesity, age, and female sex [7,8,9]. Weaker correlations are also reported between CTS and high psychological demands in the workplace associated with low decision-making authority, vibration, prolonged postures outside the neutral position of the wrist, and repetitive work [10,11]. Conversely, the use of computers is not related to an increased risk of developing CTS in the general population [12,13,14,15,16,17,18,19,20].

Clinically, CTS is characterized by multiple alterations, both motor and sensory [21]: nocturnal paraesthesia, numbness, tingling, decreased sensitivity and pain in the territory innervated by the median nerve, decreased grip strength, and atrophy of the thenar eminence [22]. Recent evidence defines CTS as a complex pain syndrome characterized by the presence of clinical, physical, neurophysiological, and psychological factors involving peripheral and central sensitization processes [23,24,25,26].

In most cases, the diagnosis of CTS occurs through anamnesis and clinical examination that makes use of the administration of various provocative tests such as the wrist flexion test (Phalen Test), the nerve percussion test (Tinel sign), the Carpal Compression Test, the Two Point Discrimination Test, and the Semmes-Weinstein Monofilament Test (SWMT) [27,28,29,30,31,32,33].

CTS is managed both conservatively and surgically [34]. The conservative treatment of CTS, like all neuro-musculoskeletal pathologies, is the responsibility of the physiotherapist (PT) [35]. With the aim of resolving the symptoms by reducing the course of this disorder, it is very important that the PT is updated according to the most recent and methodologically robust evidence of efficacy in the literature and puts in place such practices to direct own work and propose to the patient the best, most appropriate, and efficient and effective treatment.

In this regard, it is important for the PT to know which activities and occupations increase the risk and the perpetuation of the disease [18], the best specific tests for evaluation [29,33], the most effective treatments, and when medical consultation is essential [36]. While for other pathologies numerous surveys have been conducted on clinical practice in order to determine if and how it is influenced by the evidences present in the literature [37,38,39,40], for CTS in Italy have been investigated only the management and treatment following decompression of the nerve by sectioning the transverse carpus ligament [41].

No previous study has ever investigated the current practice of Italian PTs regarding the evaluation and treatment of patients with CTS. Thus, the purpose of this survey is to investigate the knowledge, management, and clinical practice of Italian PTs concerning patients with CTS in order to determine if and how the aforementioned practice is affected by current evidence of efficacy and compare it with the practice of others countries. The secondary objective is to generate inferences between the answers of the survey participants (a) the level of education, (b) the place where the professional activity takes place, and (c) the direct experience in the field calculated as the number of patients with CTS/year.

The authors of this study hypothesize a priori that PTs with a university Master’s degree, who carry out their activities in structures specialized in hand rehabilitation and who annually manage multiple cases of CTS, can answer the questions more correctly and are therefore more adherent to current evidence of efficacy.

## 2. Materials and Methods

### 2.1. Study Design

A national cross-sectional survey was carried out on the web for which the guidelines of the Strengthening the Reporting of OBservational Studies in Epidemiology (STROBE) [42] and the Checklist for Reporting Results of Internet E-Surveys (CHERRIES) were used for the construction of the Survey [42,43]. The survey was approved by the ethics committee of Lecce, Italy, with Minutes No. 40 of 10 December 2019.

### 2.2. Participants and Settings

The inclusion criterion for completing the questionnaire was to be PTs practicing on Italian territory at the time of completing the survey. Surveys with unanswered were excluded.

Potential participants were reached through a convenience sample with different ways: dissemination of the survey compilation link through social networks (Facebook, Linkedin, Twitter, Instagram), professional sharing platforms (ResearchGate), messaging services (WhatsApp, Messenger, Telegram), and e-mail. Two reminders to solicit the completion of the survey were produced and were sent by the Manual Therapy and Musculoskeletal Rehabilitation Group-GTM and by the National Association of Physiotherapists-AIFI to all its members.

In the current survey, the authors’ approach was to get the maximum number of responses in a defined period of time. This approach reflects other similar surveys currently present in the literature and conducted in Italy and Europe [44,45,46]. The time required for completing the questionnaire is 10 to 15 min. This timeframe is chosen for optimizing the response rate in the online questionnaires [47]. It was specified to possible respondents in the initial instructions. To prevent a single respondent from completing the questionnaire multiple times, the server was set up to prevent the same IP address from accessing the survey compilation more than once.

### 2.3. Development of the Questionnaire

The online cross-sectional survey was developed by the authors after a careful analysis of the most recent and methodologically robust literature concerning (a) general knowledge of the pathology of CTS, (b) diagnostic tools, (c) therapeutic techniques, and (d) outcome measures.

After the evaluation of the studies in the literature by four authors (VS, FBo, FB, SM), independently of each other, a provisional version was produced in agreement between the authors themselves. To increase the face validity of this version, the questionnaire was subjected to 10 clinicians of various experience who, blinded to each other, highlighted changes to be made to increase the clarity of the content of the questions, the thesaurus, the order of questions and answers, and the entire survey structure. The authors modified the questionnaire in this sense and proposed this version to the 10 clinicians in the plenary to judge their satisfaction with the changes. When full agreement was reached, the final version of the questionnaire was produced which was used for administration in this survey.

This final version consists of 24 questions in total. In the first part of the questionnaire 8 multiple choice questions were asked with four choices (questions 1–8 in Appendix A) regarding demographic information (e.g., age, region of origin, place of work, etc.). The goal of these questions was to better characterize the background of the professionals who had access to the survey. The subsequent 16 questions (questions 9–24 in Appendix A) were constructed to probe the knowledge of the pathology in question and are multiple choice questions with four possible answers, one of which is correct. The entire survey is available in the Appendix A. The option to decline to answer specific questions or to leave the entire questionnaire blank was also provided [48]. Participants were able to review or change effects using a back button until the end of the questionnaire.

### 2.4. Data Collection

The Survey Monkey online platform (Palo Alto, CA, USA; www.surveymonkey.com, accessed on 10 January 2021) was used. This survey was administered from December 2019 until February 2020 inclusive. This time frame is judged by the authors to be adequate because it is used in other surveys present in the literature [44,45,49]; moreover, after that date no request to fill in the questionnaire was received.

On the first access page to the survey, the purpose of the study and the promoters were specified. It was also specified that the answers would be anonymous and the informed consent was implicit through the will to fill in the survey, detected by the answer “OK” to the phrase “Do you want to proceed to participate in the survey?” (without the need for a written consent form). Participation in the same survey was voluntary and no incentives were given to increase the compilation rate.

For the data analysis, the answers were downloaded and reported in an Excel file with the data extraction method of Survey Monkey from the statistician [22] who was the only one to see, sort, and analyze the data -without the IP address- to maintain privacy of respondents [43].

### 2.5. Data Analysis

Descriptive statistics were computed in order to describe the characteristics of the sample. More specifically, mean ± standard deviation (SD) and frequency with relative percentage were calculated for intervallic and categorical variables, respectively.

In order to study any difference between subgroups of the sample (according to their academic degrees, workplaces, number of patients with CTS/year) and their response to the questionnaire, a Chi-squared independence test or Fisher’s exact test (if cell size were below 5) was run. In case the Chi-squared test revealed statistically significant differences (*p* < 0.05), adjusted standardized residuals [50] with their Bonferroni-corrected *p*-value were calculated to determinate which cells of contingency table contributed most to the significant effect [51,52]. The α level was set at *p*-value < 0.05 for all comparisons, and all statistical analyses were run with SPSS software (SPSS. Version 20 for Windows; Release 13.0.1. SPSS Inc., Chicago, IL, USA, 2004).

## 3. Results

### 3.1. Subjects

A total of 508 participants completed the survey. Most respondents (*n* = 225/508; 44.3%) are under 29 years old and female (*n* = 256/508; 50.4%) and have been working as PTs for less than 5 years (*n* = 213/508; 41.9%). Furthermore, most of the respondents only have a Bachelor’s degree in physiotherapy (279/508; 54.9%), work in hospitals (411/508; 80.9%), and come mainly from Northern Italy (246/508; 48.4%). The main field of work of the respondents involved in this study is the musculoskeletal sector (389/508; 76.6%). 334 respondents (65.7%) see 1 to 5 cases of carpal tunnel problems in one year; 106 respondents (20.9%) see 6 to 10 patients with carpal tunnel problems every year; 35 respondents (6.9%) see 11 to 15 cases; and, 30 respondents (5.9%) visit more than 15 patients with problems related to the CTS per year. Further and detailed information on the personal data of the sample included in this study can be consulted in Table 1.

### 3.2. Primary Outcome

Question 09: 92.3% of the participants (469/508) answered correctly stating that CTS is a neurological condition caused by compression of the median nerve due to an increase in pressure in the carpal canal [17,53]. In contrast, 7.7% (39/508) claimed that CTS is caused by compression of the ulnar nerve, affects the flexor muscle tendons due to functional overload or affects Guyon’s canal.

Question 10: 89.6% of the participants (455/508) correctly answered that the most likely cause of CTS is the reduction of space in the carpal canal [17,53]. A small percentage, on the other hand, replied that the most probable causes were a compression of the Guyon canal, an alteration of the collagen and a scaphoid osteophyte (10.2%, 52/508). One respondent did not answer the question (1/508, 0.2%).

Question 11: For 70.9% (360/508) of respondents, CTS patients are the responsibility of the PT [35,54,55,56]; 0.8% (4/508) replied that the patient with CTS is not the responsibility of the PT, while 28.3% (144/508) replied that CTS is the responsibility of the PT only after electromyography or a specialist medical examination.

Question 12: 88.2% (448/508) of participants answered correctly, arguing that the most common risk factors for developing CTS are female gender, obesity, diabetes, and pregnancy [8,17,53]. 11.6% (59/508) replied that age, male gender, alcohol, smoking, and hormonal diseases are the main risk factors for the development of CTS. One participant (0.2%) did not answer the question.

Question 13: 58.7% (298/508) of respondents answered that there is a correlation between computer use and CTS [57,58]. 17.5% of respondents (89/508) answered correctly that there is no correlation between computer use and CTS, 16.3% (83/508) argue that the correlation exists, but only if computer use it is prolonged for more than 10 h a day, while 7.3% (37/508) of the respondents support the correlation, but only if the use of the computer is accompanied by the use of a non-ergonomic mouse. One participant did not answer the question (1/508, 0.2%).

Question 14: 81.7% of participants (415/508) answered correctly, claiming that CTS is characterized by impaired sensitivity, tingling, and numbness in the first three fingers of the hand [59,60]. 15.7% (80/508) argued that impaired sensation, tingling, and numbness of the first three fingers characterize CTS, while 2.6% (13/508) argued that joint limitation of the radio-carpica or hypothenary muscle strength deficit may characterize CTS.

Question 15: 74.8% (380/508) of the participants answered correctly, indicating the “hypotrophy of the thenar eminence” [60]. 15.9% (81/508) of the respondents indicated “dorsal and volar hypotrophy of the hand and hypotrophy of the thenar and hypothenar eminences, 6.1% (31/508) indicated “hypotrophy of the hypothenar eminence” and 2.8% (14/508) indicated “localized edema in the distal joints”. Two respondents (0.4%) did not answer the question.

Question 16: 52.0% (264/508) of the participants replied that the use of Semmes-Weinstein Monofilaments is indicated [27,54,59,61]. 17.7% (90/508) replied that the use of a pin is recommended, 15.0% (76/508) believed that there was no appropriate tool for assessing sensitivity, while 14.4% (73/508) argued that the description of the symptoms by the patient is sufficient.

Question 17: 67.1% (341/508) of participants answered correctly, supporting the use of the Wrist Flexion Test (Phalen), the Nerve Percussion Test (Tinel), the Functional Dexterity test and the two points discrimination test [27,54,59,61]. 17.9% (91/508) of respondents support the choice of the cluster formed by the Phalen Test, Upper Limb Neurodynamic Test 1 (ULNT-1) and stability test of the scaphoid compared to the other carpal bones (Watson’s Test), 6.9% (35/508) supports the choice of the cluster formed by the Cozen’s test, a sensitive evaluation test in the thenar area and in the palm of the hand, while 6.5% (33/508) supports the use of the cluster formed by the Phalen Test, Upper Limb Neurodynamic Test 3 (ULNT-3 for the ulnar nerve), two points discrimination test. 1.6% (8/508) of respondents did not answer this question.

Question 18: 83.3% (423/508) of respondents respond as recommended by the literature [54,61] indicating “evaluation of sensitivity, dexterity, strength, pain together with a questionnaire for the evaluation of symptoms and functionality”. 9.8% (50/508) responded by supporting the use of “Administration of Visual Analogue Scale (VAS) or the Numeric Pain Rating Scale (NPRS)” together with measuring force with dynamometer and manual dexterity: 6.3% (32/508) of the respondents argues that there is only a need for an interview with the patient. Finally, 0.6% (3/508) of the respondents did not answer the question.

Question 19: 53.0% (269/508) of respondents answered correctly, claiming that they would recommend or make an orthotic for the management of patients with CTS [54]. 44.3% (225/508) would not recommend it, or would do it only in the case of concomitant rhizo-arthrosis; finally, 2.4% (12/508) argue that the use of the orthotic is contraindicated in patients with CTS. 0.4% of respondents (2/508) did not answer the question.

Question 20: 56.1% of respondents (285/508) answered that they do not use instrumental therapies in their clinical practice because these methods are supported by weak or moderate evidence of efficacy [54]. 38.2 (194/508) replied that the evidence of efficacy in support of physical therapies in the treatment of the patient with CTS is weak or moderate, but still uses these methods in clinical practice. 4.7% of respondents (24/508) argue that evidence of efficacy in support of physical therapies is strong: 3.7% (19/508) use them in their clinical practice, while 1% (5/508) of respondents however does not use them. 1% (5/508) of respondents did not answer the question.

Question 21: 49.4% of respondents (251/508) answered correctly, arguing that the evidence of efficacy in support of nerve and tendon glide techniques is limited/moderate [62,63], but they still use these techniques in their own clinical practice. 34.4% (175/508) of respondents responded that the evidence is strong and that they use these techniques in their clinical practice; 7.5% (38/508) of the respondents replied that the evidence is limited/moderate and for this reason I do not use these techniques while 6.9% (35/508) of the respondents stated that even if based on strong evidence of efficacy, these techniques do not are used in their own clinical practice. Finally, 1.8% (9/508) of the respondents did not answer the question. The percentage values of use of the main intervention strategies are presented in Figure 1.

Question 22: 90% of respondents (457/508) answered correctly by choosing “education, manual therapy, myofascial therapy and therapeutic exercise” [54,64,65], while 9.8% (50/508) of the respondents used massotherapy, physical therapy, joint mobilization of the radius and stretching. 0.2% (1/508) did not answer the question (Figure 2).

Question 23: 71.7% of respondents (364/508) answered correctly by choosing “Yes and I adapt my clinical practice accordingly” [24,25,66,67], while 23.4% (119/508) of the respondents maintain that psychosocial factors can influence the patient’s outcomes but do not know how to adapt their clinical practice and limit themselves to treating the aspects of education and explanation of the central sensitization processes. 4.5% (23/508) of the respondents instead argued that psychosocial factors do not influence the outcomes of patients with CTS. 0.4% (2/508) did not answer the question.

Question 24: 92.1% of respondents (468/508) answered correctly, arguing that the surgical approach should be contemplated in case of failure of conservative treatment [36,54], while 7.7% (39/508) believe that the surgical approach should not be contemplated or only in concomitant stenosing synovitis. 0.2% (1/508) did not answer the question.

The detailed answers for each question, provided by survey respondents, are described in Appendix B.

### 3.3. Secondary Outcome

The results investigating these topics are summarized in Table 2, Table 3 and Table 4.

For the inference between the correct answer and the respondents’ level of education, the answers are summarized in Table 2 and inferences with statistical significance is presented in Figure 3.

The results of the inferences between the correct answer to each question and the respondents’ prevailing workplace are summarized in Table 3 and inference with statistical significance is graphically presented in Figure 4.

## 4. Discussion

The objectives of this study are to provide information on the knowledge and current clinical practice of Italian PTs in relation to CTS, to determine whether this practice is in line with current evidence of efficacy, and to evaluate the inferences between participants’ responses to the survey and the level of education, workplace, and direct experience in the field (calculated with number of CTS patients/year).

The definition that most respondents in this study believe correct for CTS is in line with the American Academy of Orthopedic Surgeons (AAOS) [53], which describes it as “symptomatic neuropathy from compression of the median nerve at the wrist level, physiologically characterized by the presence of an increase in pressure within the carpal canal and a reduction in nerve function at this level”.

Almost all of the PTs interviewed know the main risk factors reported in the literature for the development of CTS, such as female gender, obesity, diabetes, and pregnancy [8,17,61]. A widespread belief among Italian PTs is that CTS is associated with the use of computers, although the evidence available in the literature for this claim is controversial [14,18]. Some studies suggest that excessive computer use may be a minor risk factor for the onset of CTS [61], but two recent meta-analyzes have concluded that to date it is not possible to establish a direct causal association with certainty [57,58].

Most Italian PTs seem to be aware of the main nosological features of CTS such as pain, numbness, tingling, paraesthesia, and impaired sensitivity in the distribution area of the median nerve [59]. In addition, more than three quarters of the interviewed sample appear to recognize typical alterations of the most advanced stage such as hypotrophy or atrophy of the thenar eminence, with a reduction in grip strength and manual dexterity and consequent loss of function [60]. This finding is in line with Australian PTs’ confidence in recognizing neurological disorders [68].

The evidence currently available in the literature suggests that the physical examination should include a battery of diagnostic tests, as well as specific outcome measures [54]; in fact, no single test alone can be sufficient for a definitive diagnosis of CTS [17]. Provocative tests considered valid for the diagnosis of CTS and commonly used internationally and by more than half of the sample interviewed include Tinel’s sign, Phalen’s test, Dellon-modified Moberg pick-up test or Purdue Pegboard, the two-points static discrimination test, and the Semmes-Weinstein monofilament test (SWMT) [27,54,59].

The SWMT is a valid test known only by just over half of Italian PTs and is a quantitative and objective method for mapping the loss of sensation within the distribution of the median nerve of patients with moderate to severe CTS [54,69]. Unfortunately, it emerges from this study that most Italian PTs relegate the sensitivity test to instruments that have proved unreliable (e.g., pin) or do not perform it on account of an inability to identify any appropriate instrument.

Outcome measures, known to most of the PTs who participated in the survey, include a pain measurement scale, measurement of global and clamp force through the use of a dynamometer, assessment of manual dexterity, assessment sensitivity, and the administration of validated scales for the evaluation of symptoms and function (e.g., PROMS validated in Italian such as Disabilities of the Arm, Shoulder and Hand questionnaire [DASH] or Boston Carpal Tunnel Syndrome Questionnaire [BCTQ]) [54,61].

With CTS being a musculoskeletal disorder that can be treated conservatively and according to rehabilitative procedures, conservative treatment of CTS is the responsibility of the PT [35]. Additionally, the conservative approach has shown success in many studies in the literature with various follow-ups [54,55,56]. However, about one third of the respondents to this questionnaire would treat the patient with CTS only after PT has undergone a specialist medical examination or a median nerve conduction study with electromyography (EMG). This result is not perfectly in line with the evidence of efficacy present in the literature, which instead tend to promote a medical-surgical consultation only when patients regress, do not improve with conservative management, or present severe CTS and severe atrophy of the thenar eminence [54]. Electrodiagnostic studies and imaging studies, on the other hand, are generally reserved for patients in whom diagnostic certainty is questionable, but in the case of CTS there is controversy in the literature: although EMG is commonly used, there is limited evidence regarding the its usefulness in diagnosing CTS [17,53,70]. In fact, with a sensitivity from 49% to 84% and a specificity from 95% to 99%, it may not be sufficiently reliable, when negative, to exclude the presence of pathology [53]. Recent studies have shown that sonoelastography is a useful non-invasive and promising modality to diagnose CTS [71]. As for the conservative treatment of CTS, the best evidence available in the literature supports the use of a night orthotic with the wrist in a neutral position with the aim of reducing symptoms [54]. Only half of the recruited sample proposes the orthotic as a treatment strategy for CTS. These results are different from those of an American survey, which shows that the orthotic is the most commonly used conservative intervention by all PTs who are members of the American Society of Hand [72]. The low use of orthoses in this survey may be related to the fact that PTs are not comfortable with this intervention. Probably, if the sample consisted exclusively of hand therapists the percentage of orthotic use would be higher and therefore the results would be more similar to the American survey [72].

The 2019 JOSPT (Journal of Orthopedic & Sports Physical Therapy) guideline showed that instrumental therapies such as magnetotherapy, low-level laser therapy, ultrasound, and iontophoresis provide no benefit in the management of CTS [54]. This survey highlights a substantial confusion in the use of instrumental therapy in the clinical practice of Italian PTs who are almost equally divided between those who use it and those who do not. These results are in contrast to the US survey, which shows that instrumental therapies, with the exception of ultrasound, are scarcely used by American PTs [72]. Probably the tendency of Italian PTs to propose mechanically ineffective interventions in clinical practice derives from the fact that, as also highlighted in the study by Giovannico et al. [73], basic academic training programs would need to be modified in order to allow this scientific discipline to reach its maximum potential.

Nerve mobilization techniques and tendon glide exercises appear to be widely used by Italian PTs, despite conflicting and limited evidence of efficacy supporting these interventions [62,63,74,75,76]. The results of the present study are similar to both those of the American survey by Parish et al. [72] and those of the worldwide survey on the clinical practice of PTs and occupational therapists following the Carpal Tunnel Release (CTR) [41].

The 2019 JOSPT guideline suggests that patient education programs, ergonomic interventions, manual therapy of the cervical spine and upper limb, myofascial and therapeutic exercise can be offered to patients with CTS [54], although these interventions are supported by moderate or limited and low quality evidence [54,64,65]. The results of this survey reflect current recommendations and are similar to those reported in the US survey [72]. In fact, almost all of the respondents choose to provide such a combination of interventions for the management of the CTS.

Although it has now been shown in the literature that the painful function and experience of patients with CTS can be influenced by psychosocial factors [24,25,26,66,67,77], there is an unfortunate tendency to overlook these factors in clinical practice. The present survey reflects this problem so much that almost a quarter of the participants, although aware of the importance of psychosocial variables, seem to find it difficult to identify and manage these variables.

Finally, almost all respondents agree that, in case of failure of conservative treatment, the surgical approach can be a solution for patients with CTS [36,54]. This is also confirmed by another survey that identifies the perception of the benefit of surgery by hand therapists, which appears to be similar to that of surgeons [78].

As expected by the authors of this study, some significant differences emerged from the analysis of the results regarding the cultural level of the PTs interviewed: those specialized in holding a University Master’s degree seem to be more in line with the current evidence of effectiveness. The results of this study align with those of other Italian surveys [46] and may be justified by the fact that in Italy the Master’s Degree Course provides professional training mainly oriented towards the acquisition of skills in management, training, and research processes and not towards the deepening of specialist clinical knowledge.

Contrary to what was expected in the initial hypotheses, however, no differences were highlighted in relation to the place of professional activity or the clinical experience of the Italian PTs. This means that carrying out one’s activity in a specialized setting or managing multiple CTS cases annually does not necessarily imply greater knowledge, competence, and adherence to evidence-based clinical practice (EBP), but rather, it is probably the study—and continuous individual study—of the subject that allows the clinician to make decisions based on evidence.

### Strengths and Limitations of the Study

This study is limited to Italian PTs only and analyzes only a part of the population of clinical PTs: those who usually use social media or electronic devices (even if now almost all healthcare professionals are familiar with technology and the use of smartphones as it has become common practice in the management of health communication) [79]. This may not represent the entire PT population and thus represent a selection bias. This limitation has been identified in other surveys in the musculoskeletal field [38]. In fact, it is recognized that self-administered questionnaires are not useful for studying populations that do not use the technology [80]. Furthermore, the choice of orienting the survey towards a single professional category may not represent the interprofessional idea of the evaluation and treatment of CTS. This is the first survey aimed at analyzing the perception of PTs on the subject.

The strength of this study is in the rigorous statistical analysis, the analysis of the sub-classification of the sample, and the methodological construction of the survey. This survey is also the result of a scrupulous and rigorous research of the literature and can be the starting point for monitoring the knowledge on this topic in the coming years. This study is the first to analyze the knowledge and management of pathology by PTs with different specializations and not only hand rehabilitation experts as has happened up to now. This survey is also the first study undertaken not only in the Italian context, but also in the European context, and thus could be the starting point for studies in/of other nations.

## 5. Conclusions

Most respondents are aware of the nosological characteristics, major risk factors, diagnostic tests, and specific outcome measures of CTS. The management of the disease was not always, and not completely, in line with current evidence. The survey showed greater adherence on the part of PTs holding a Master’s degree. Interventions such as education, manual therapy, and therapeutic exercise, as well as the nerve glide and tendon techniques are widely used, while the orthotic is only offered by half of the sample.

## Figures and Tables

**Figure 1 ijerph-18-01995-f001:**
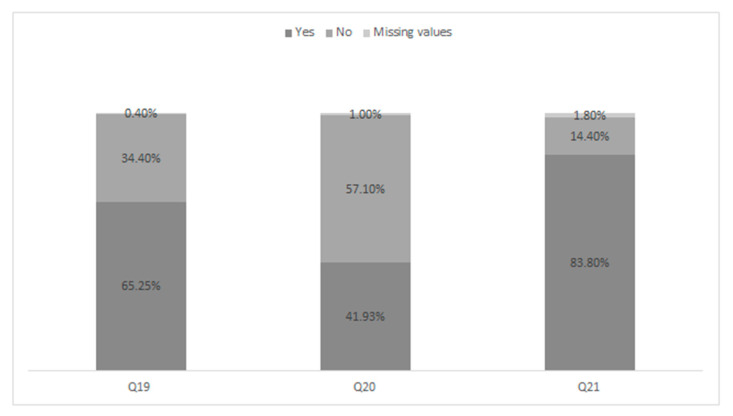
Clinical management for orthotics, instrumental therapies and neuro/tendons glides techniques. Tabled as percentage of respondent (*n* = 508). Q: question. Q19: Orthotic; Q20: Instrumental Therapies; Q21: Neurodynamic/tendons glides techniques.

**Figure 2 ijerph-18-01995-f002:**
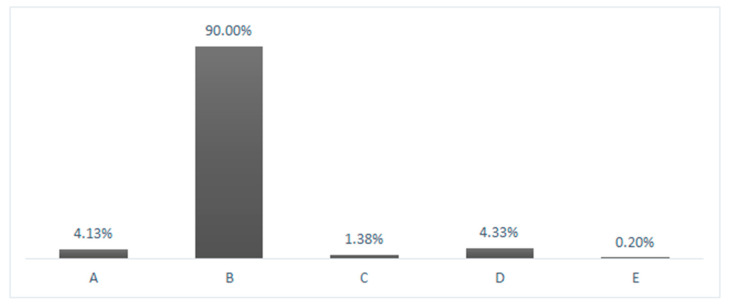
Most frequent treatment strategy for carpal tunnel syndrome. A: Massage therapy, instrumental therapy; B: Education, manual therapy, myofascial therapy, therapeutic exercise; C: Joint mobilization of the radiocarpal joint, stretching; D: None of the previous answers; E: Missing values.

**Figure 3 ijerph-18-01995-f003:**
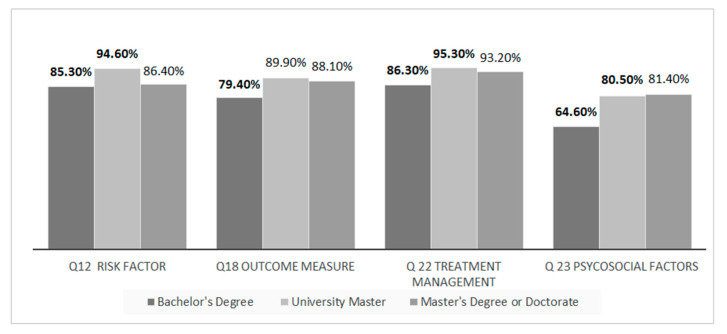
Inferences between educational qualifications and correct answers (Q12, Q18, Q22, Q23). Acronyms: Q: question. Bold values outside the box are the ones with statistical significance in the residual *p*-values. Tabled as percentage of respondent (*n* = 508).

**Figure 4 ijerph-18-01995-f004:**
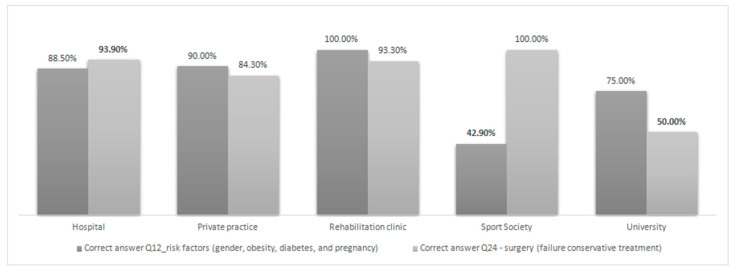
Inferences between places of professional activity and correct answers (Q12 and Q24). Acronyms: Q, question. Bold values outside the box are the ones with statistical significance in the residual *p*-values. Tabled as percentage of respondent (*n* = 508).

**Table 1 ijerph-18-01995-t001:** Demographic characteristics of respondents.

		Answers (*n*)	Percentage (%)	Missing (*n*)
Q 1-Age	<29	231	44.85	0
	30–39	167	32.43	
	40–49	66	12.82	
	≥50	51	9.90	
Q 2-Gender	Female	260	50.49	0
	Male	255	49.51	
Q 3-Years of experience	<5 years	219	42.52	0
	5–10 years	110	21.36	
	11–20 years	123	23.88	
	>20 years	63	12.23	
Q 4-University degree	Three-year degree	279	54.28	1
	University Master	169	32.88	
	Master’s degree	56	10.89	
	Doctorate	3	0.58	
Q 5-Where do you do business	Public body (hospital)	72	14.01	1
	Private body	412	80.16	
	Wrist/hand specialized body	15	2.92	
	Society	8	1.56	
	University	7	1.36	
Q 6-Business area	Skeletal muscle	391	76.22	2
	Geriatric	49	9.55	
	Neurological	59	11.50	
	Other	14	2.73	
Q 7-Region of origin	Northern Italy	247	47.96	0
	Southern Italy	115	22.33	
	Center of Italy	153	29.71	
Q 8-Number of cases per year	1–5	341	66.60	3
	6–10	106	20.70	
	11–15	35	6.84	
	>15	30	5.86	

Acronyms: Q, question.

**Table 2 ijerph-18-01995-t002:** Inferences between correct answers and the respondents’ educational qualifications.

	Educational Qualifications			
Correct Answer per Question	Bachelor’s Degree	University Master	Master’s Degree or Doctorate	*p* Values
Q9—CTS is a neurological condition caused by compression of the median nerve due to an increase in pressure within the carpal tunnel	88.5% (247/508)	98.2% (166/508)	93.2% (55/508)	0.001 *
Adjusted residuals	3.50	3.50	0.30	
Residual’s *p*-values (Bonferroni *p*-values = 0.008)	**0.0004**	**0.0004**	0.7641	
Q10—CTS is caused by reduction of space within the carpal canal	87.1% (242/508)	93.5% (158/508)	91.5% (54/508)	0.084
Q11—Patients with CTS are responsibility of the physiotherapist	63.0% (175/508)	80.5% (136/508)	81.4% (48/508)	0.001 *
Adjusted residuals	4.43	3.38	1.90	
Residual’s *p*-values (Bonferroni *p*-values = 0.0055)	**0.0000**	**0.0007**	0.0574	
Q12—Female gender, obesity, diabetes and pregnancy are risk factors for CTS	85.3% (238/508)	94.6% (159/508)	86.4% (51/508)	0.010 *
Adjusted residuals	2.53	3.04	0.54	
Residual’s *p*-values (Bonferroni *p*-values = 0.0083)	0.0114	**0.0023**	0.58919	
Q13—There’s no association between CTS and computer	61.9% (172/508)	52.1% (88/508)	62.7% (37/508)	0.088
Q14—Altered sensitivity, tingling and numbness of the first three fingers are the main characters of CTS	80.3% (224/508)	83.4% (141/508)	83.1% (49/508)	0.843
Q15—In patients with CTS is possible to find hypotrophy of the thenar eminence	71.2% (198/508)	81.0% (136/508)	78% (46/508)	0.150
Q16—The Semmes-Weinstein Monofilaments are the best tool for the tactile sensitivity examination	35.3% (97/508)	77.4% (130/508)	61% (36/508)	0.000 *
Adjusted residuals	8.47	7.95	1.41	
Residual’s *p*-values (Bonferroni *p*-values = 0.0041)	**0.0000**	**0.0000**	0.15853	
Q17—Wrist flexion test (Phalen’s test), nerve percussion test (Tinel’s sign), Functional Dexterity test and two-point discrimination are most used clinical test	66.3% (181/508)	71.4% (120/508)	67.2% (39/508)	0.068
Q18—Measurement of strength with dynamometer and of sensitivity, manual dexterity, strength and pain and administration of a questionnaire for the evaluation of symptoms and function are the most used outcome evaluation tools	79.4% (220/508)	89.9% (151/508)	88.1% (52/508)	0.010 *
Adjusted residuals	3.04	2.57	0.94	
Residual’s *p*-values (Bonferroni *p*-values = 0.0055)	**0.00236**	0.01016	0.34721	
Q19—I advice or build an orthotic	50.9% (142/508)	57.5% (96/508)	50.8% (30/508)	0.073
Q20—I don’t use instrumental therapies in my clinical practice; supporting evidences are weak/moderate	52.6% (144/508)	61.5% (104/508)	61% (36/508)	0.145
Q21—There is limited evidence on neural and tendon glide techniques and that’s why I don’t use it in my clinical practice	47.1% (128/508)	52.7% (88/508)	57.6% (34/508)	0.152
Q22—Education, manual therapy, myofascial therapy, therapeutic exercise are most used treatment strategies	86.3% (240/508)	95.3% (161/508)	93.2% (55/508)	0.006 *
Adjusted residuals	3.15	2.75	0.85	
Residual’s *p*-values (Bonferroni *p*-values = 0.0083)	**0.00163**	**0.00595**	0.39532	
Q23—I adapt my clinical practice accordingly with the influence of psychosocial factors on the patient outcome	64.6% (179/508)	80.5% (136/508)	81.4% (48/508)	0.002 *
Adjusted residuals	4.00	3.05	2.63	
Residual’s *p*-values (Bonferroni *p*-values = 0.0055)	**0.00006**	**0.00228**	0.08543	
Q24—the surgical approach can be a solution in cases of failure of conservative treatment (persistence of symptoms)	90.6% (252/508)	96.4% (163/508)	88.1% (52/508)	0.037 *
Adjusted residuals	1.53	2.48	1.27	
Residual’s *p*-values (Bonferroni *p*-values = 0.00833)	0.12601	0.01313	0.20408	

Acronyms: CTS, Carpal Tunnel Syndrome; n.s., not-significant; Q, question. NOTE: * = Significant *p*-values (*p* < 0.05); statistically significant differences according to the corrected residuals are in **bold**.

**Table 3 ijerph-18-01995-t003:** Inferences between correct answers and place of professional activity of the respondents.

	Place of Professional Activity					
Correct Answer per Question	Hospital	Private Practice	Rehabilitation Clinic	Sport Society	University	*p* Value
Q9—CTS is a neurological condition caused by compression of the median nerve due to an increase in pressure within the carpal tunnel	93.4% (384/508)	87.1% (61/50)	100% (15/508)	71.4% (5/508)	100% (4/508)	0.051
Q10—CTS is caused by reduction of space within the carpal canal	91.0% (373/508)	81.4% (57/50)	100% (15/508)	85.7% (6/508)	75.0% (3/508)	0.069
Q11—Patients with CTS are responsibility of the physiotherapist	71.0% (292/508)	72.9% (51/50)	66.7% (10/508)	57.1% (4/508)	75.0% (3/508)	0.983
Q12—Female gender, obesity, diabetes and pregnancy are risk factors for CTS	88.5% (363/508)	90.0% (63/50)	100.0% (15/508)	42.9% (3/508)	75% (3/508)	0.002 *
Adjusted residuals	0.28	0.47	1.43	3.78	0.83	
Residual’s *p*-values (Bonferroni *p*-value = 0.005)	0.77947	0.63835	0.15271	**0.00015**	0.40653	
Q13—There’s no association between CTS and computer	57.6% (236/508)	67.1% (47/50)	46.7% (7/508)	57.1% (4/508)	100% (4/508)	0.699
Q14—Altered sensitivity, tingling and numbness of the first three fingers are the main characters of CTS	83.0% (341/508)	72.9% (51/50)	86.7% (13/508)	71.4% (5/508)	100% (4/508)	0.462
Q15—In patients with CTS is possible to find hypotrophy of the thenar eminence	76.3% (312/508)	68.6% (48/50)	86.7% (13/508)	42.9% (3/508)	100% (4/508)	0.596
Q16—The Semmes-Weinstein Monofilaments are the best tool for the tactile sensitivity examination	52.2% (212/508)	47.1% (33/50)	86.7% (3/508)	57.1% (4/508)	50% (2/508)	0.252
Q17—Wrist flexion test (Phalen’s test), nerve percussion test (Tinel’s sign), Functional Dexterity test and two-point discrimination are most used clinical test	67.2% (272/508)	67.6% (46/50)	93.3% (14/508)	71.4% (5/508)	75% (3/508)	0.229
Q18—Measurement of strength with dynamometer and of sensitivity, manual dexterity, strength and pain and administration of a questionnaire for the evaluation of symptoms and function are the most used outcome evaluation tools	84.1% (345/508)	82.4% (56/50)	86.7% (13/508)	71.4% (5/508)	100% (4/508)	0.836
Q19—I advice or build an orthotic	53.2% (218/508)	50.7% (35/50)	80.0% (12/508)	28.6% (2/508)	25% (1/508)	0.123
Q20—I don’t use instrumental therapies in my clinical practice; supporting evidences are weak/moderate	58.5% (238/508)	53.6% (37/50)	33.3% (5/508)	42.9% (3/508)	50% (2/508)	0.612
Q21—There is limited evidence on nerve and tendon glide techniques and that’s why I don’t use it in my clinical practice	50.2% (203/508)	50% (34/50)	53.3% (8/508)	42.9% (3/508)	75% (3/508)	0.900
Q22—Education, manual therapy, myofascial therapy, therapeutic exercise are most used treatment strategies	90.3% (371/508)	92.8% (64/50)	80.0% (12/508)	100% (7/508)	75% (3/508)	0.390
Q23—I adapt my clinical practice accordingly with the influence of psychosocial factors on the patient outcome	71% (291/508)	75.4% (52/50)	73.3% (11/508)	100% (7/508)	75% (3/508)	0.861
Q24—the surgical approach can be a solution in cases of failure of conservative treatment (persistence of symptoms)	93.9% (385/508)	84.3% (59/50)	93.3% (14/508)	100% (7/508)	50% (2/508)	0.001 *
Adjusted residuals	2.81	2.71	0.15	0.77	3.18	
Residual’s *p*-values (Bonferroni *p*-value = 0.005)	**0.00495**	0.00672	0.88076	0.44129	**0.00147**	

Acronyms: Q, question; CTS, Carpal Tunnel Syndrome; n.s., non-significant. NOTES: * = significant *p*-values (*p* < 0.05), statistically significant differences according to the corrected residuals are in **bold****.**

**Table 4 ijerph-18-01995-t004:** Inferences between correct answers and number of cases treated each year by respondents.

	Number of CTS Patients/Year				
Correct Answer per Question	1–5	6–10	11–15	>15	*p* Value
Q9—CTS is a neurological condition caused by compression of the median nerve due to an increase in pressure within the carpal tunnel	93.4% (312/508)	91.5% (97/508)	80.0% (28/508)	96.7% (29/508)	0.031 *
Adjusted residuals	1.34	0.33	2.82	0.93	
Residual’s *p*-values (Bonferroni *p*-value = 0.0062)	0.18024	0.74139	**0.00480**	0.35237	
Q10—CTS is caused by reduction of space within the carpal canal	90.4% (302/508)	87.7% (93/508)	91.4% (32/508)	86.2% (25/508)	0.771
Q11—Patients with CTS are responsibility of the physiotherapist	72.2% (241/508)	70.8% (75/508)	60.0% (21/508)	70% (21/508)	0.784
Q12—Female gender, obesity, diabetes and pregnancy are risk factors for CTS	88.6% (295/508)	84.9% (90/508)	88.6% (31/508)	100% (30/508)	0.155
Q13—There’s no association between CTS and computer	57.2% (191/508)	56.2% (59/508)	68.6% (24/508)	70% (21/508)	0.281
Q14—Altered sensitivity, tingling and numbness of the first three fingers are the main characters of CTS	81.7% (273/508)	83% (88/508)	77.1% (27/508)	83.3% (25/508)	0.239
Q15—In patients with CTS is possible to find hypotrophy of the thenar eminence	77.2% (257/508)	74.3% (78/508)	57.1% (20/508)	76.7% (23/508)	0.308
Q16—The Semmes-Weinstein Monofilaments are the best tool for the tactile sensitivity examination	53.3% (177/508)	51.0% (53/508)	48.6% (17/508)	53.3% (16/508)	0.619
Q17—Wrist flexion test (Phalen’s test), nerve percussion test (Tinel’s sign), Functional Dexterity test and two-point discrimination are most used clinical test	64.4% (212/508)	71.4% (75/508)	77.1% (27/508)	89.7% (26/508)	0.033 *
Adjusted residuals	2.57	0.78	1.17	2.55	
Residual’s *p*-values (Bonferroni *p*-value = 0.0031)	0.01016	0.43539	0.24200	0.01077	
Q18—Measurement of strength with dynamometer and of sensitivity, manual dexterity, strength and pain and administration of a questionnaire for the evaluation of symptoms and function are the most used outcome evaluation tools	86.1% (285/508)	81.1% (86/508)	68.6% (24/508)	83.3% (25/508)	0.120
Q19—I advice or build an orthotic	55.1% (183/508)	49.1% (52/508)	45.7% (16/508)	60.0% (18/508)	0.310
Q20—I don’t use instrumental therapies in my clinical practice; supporting evidences are weak/moderate	60.6% (200/508)	52.8% (56/508)	45.7% (16/508)	40.0% (12/508)	0.060
Q21—There is limited evidence on neural and tendon glide techniques and that’s why I don’t use it in my clinical practice	50.2% (165/508)	50% (52/508)	48.6% (17/508)	53.3% (16/508)	0.612
Q22—Education, manual therapy, myofascial therapy, therapeutic exercise are most used treatment strategies	92.2% (307/508)	88.7% (94/508)	80% (28/508)	83.3% (25/508)	0.060
Q23—I adapt my clinical practice accordingly with the influence of psychosocial factors on the patient outcome	71.1% (236/508)	74.5% (79/508)	71.4% (25/508)	70.0% (21/508)	0.948
Q24—the surgical approach can be a solution in cases of failure of conservative treatment (persistence of symptoms)	90.4% (301/508)	95.3% (101/508)	97.1% (34/508)	96.7% (29/508)	0.174

Acronyms: Q, question; CTS, Carpal Tunnel Syndrome; n.s., non-significant. NOTES: * = significant *p*-values (*p* < 0.05), statistically significant differences according to the corrected residuals are in **bold**.

## Data Availability

The data presented in this study are available on request from the corresponding author.

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
