# Peer review of "Carpal Tunnel Syndrome: A National Survey to Monitor Knowledge and Operating Methods"

_ijerph, 2021, doi:10.3390/ijerph18041995_

Round 1
Reviewer 1 Report
I had a pleasure to review the paper: Carpal Tunnel Syndrome: a National Survey to Monitor Knowledge and Operating Methods. The text is interesting and written on the basis of well-chosen literature. I only have a few small suggestions that are presented below.
Table 1 in page 4 has no title. Instead it says: „This is a table. Tables should be placed in the main text near to the first time they are cited”.
Furthermore in Table 1 in page 4 the number of participants with University degree is presented with three different values (ie. 279, 169, 56) near the Three-year degree. Below, in the same table in page 5 the University Master had 3 participants and PhD none of participants. This is probably an error. Please check it.
In page 5 the Result for Question 13 shows that: 58.7% (298/508) of respondents answered correctly claiming that there is a correlation between computer use and carpal tunnel syndrome”. This is however not a correct answer as explained in Introduction in page 2, lines 45-46 where the authors state that there is no association with computer use and CTS. Please consider removing “correctly”.
In the Discussion section, page 18, when discussing CTS diagnostics, the authors could consider adding one sentence on ultrasound and its promising new modalities such as elastography.
In References citations 3 and 4 are provided in Italian and seems to be not complete.
Author Response
Dear Editor-in-Chief and reviewers,
We highly appreciated the valuable comments of the reviewers. We have made every effort to respond to their comments and suggestions. The modified text is highlighted in yellow in the revised manuscript.
Reviewer 1:
I had a pleasure to review the paper: Carpal Tunnel Syndrome: a National Survey to Monitor Knowledge and Operating Methods. The text is interesting and written on the basis of well-chosen literature. I only have a few small suggestions that are presented below.
REPLY: We thank the reviewer for your time and the suggestion to improve our manuscript.
Table 1 in page 4 has no title. Instead it says: This is a table. Tables should be placed in the main text near to the first time they are cited”.
REPLY: We are sorry for the mistake. We amended as requested.
Furthermore in Table 1 in page 4 the number of participants with University degree is presented with three different values (ie. 279, 169, 56) near the Three-year degree. Below, in the same table in page 5 the University Master had 3 participants and PhD none of participants. This is probably an error. Please check it.
REPLY: We thank the reviewer for this note. We amended.
In page 5 the Result for Question 13 shows that: 58.7% (298/508) of respondents answered correctly claiming that there is a correlation between computer use and carpal tunnel syndrome”. This is however not a correct answer as explained in Introduction in page 2, lines 45-46 where the authors state that there is no association with computer use and CTS. Please consider removing “correctly”.
REPLY: We thank the reviewer for this note. We amended in the text following your suggestion.
In the Discussion section, page 18, when discussing CTS diagnostics, the authors could consider adding one sentence on ultrasound and its promising new modalities such as elastography.
REPLY: We thank the reviewer for this suggestion aiming to optimize our manuscript. We added a sentence in discussion section as you suggested. Now we state: “Recent studies have shown that sonoelastography is a useful non-invasive and promising modality to diagnose CTS [71]”.
In References citations 3 and 4 are provided in Italian and seems to be not complete.
REPLY: We thank the reviewer . We amend the reference list as suggested.
Reviewer 2 Report
Dear authors,
I am attaching the document with the improvement suggestions for your paper. Congratulations, it's very interesting!
Best regards.
What do you want to do ? New mailCop

Author Response
Dear Editor-in-Chief and reviewers,
We highly appreciated the valuable comments of the reviewers. We have made every effort to respond to their comments and suggestions. The modified text is highlighted in yellow in the revised manuscript.
Reviewer 2:
This is an article focus on of an area with good justification and need to go deeper into the topic “Carpal Tunnel Syndrome: a National Survey to Monitor Knowledge and Operating Methods”. Overall most methods have been employed to a good standard and described well. I have a few comments and suggestions to help improve clarity in parts of the paper:
REPLY: We thank the reviewer for your time and the suggestion to improve our manuscript.
In the abstract:
- In the abstract it would be interesting to include the type of study that has been carried out and the total time that the study lasted.
REPLY: We thank the reviewer for this suggestion aiming to improve our manuscript. We amended the abstract section as you suggested.
In the material and methods:
- In addition to the inclusion criteria, it would be interesting to specify exclusion criteria, for example, if surveys where there were unanswered questions were excluded.
REPLY: thank you for this note. We added a sentence following your suggestion. Now we state: “Surveys with unanswered questions were excluded”.
- How was the sample of physiotherapists who participated in the study finally selected? Was a convenience sampling used?
REPLY: We thank the reviewer for the suggestion. We recruited the participants through a convenience sample. We added this information in the Methods section. Now we state: “Potential participants were reached through a convenience sample with different ways…”
In the results:
- The font size is too small in figure 2 and makes reading difficult.
REPLY: We thank the reviewer for the suggestion. We amended Figure 2
- The font size in table 2 could be made a little smaller so that the table would be shorter, since this paper is very long.
REPLY: We thank the reviewer for the suggestion. We amended Figure 2
- The font size is too small in figure 4 and makes reading difficult.
REPLY: We thank the reviewer for the suggestion. We amended Figure 4
In the conclusion:
- The first sentence in the conclusions is about repetitive information regarding the strengths of the study. So it could be deleted, no need to include repetitive information.
REPLY: We thank the reviewer for the suggestion. We deleted that sentence as you suggested
In general:
- It would be more suitable to write in the text with impersonal form: “our” lines 72, 375, 376, 405, 438, 446
REPLY: We thank the reviewer for the suggestion. We amended as requested